# Reducing Driving Risk Factors in Adolescents with Attention Deficit Hyperactivity Disorder (ADHD): Insights from EEG and Eye-Tracking Analysis

**DOI:** 10.3390/s24113319

**Published:** 2024-05-23

**Authors:** Anat Keren, Orit Fisher, Anwar Hamde, Shlomit Tsafrir, Navah Z. Ratzon

**Affiliations:** 1Department of Occupational Therapy, Tel Aviv University, Tel Aviv 6997801, Israel; anatk87@gmail.com (A.K.); orit.fisher@mail.huji.ac.il (O.F.); anwarh5790@gmail.com (A.H.); 2The Child and Adolescent Psychiatry Division, Edmond and Lily Safra Children’s Hospital, Sheba Medical Center, Tel Hashomer, Ramat Gan 5262000, Israel; shlomit.tsafrir@sheba.health.gov.il; 3The Faculty of Medicine & Health Professions, Tel Aviv University, Tel Aviv 6997801, Israel

**Keywords:** ADHD, driving stimulation, intervention program

## Abstract

Adolescents with attention deficit hyperactivity disorder (ADHD) face significant driving challenges due to deficits in attention and executive functioning, elevating their road risks. Previous interventions targeting driving safety among this cohort have typically addressed isolated aspects (e.g., cognitive or behavioral factors) or relied on uniform solutions. However, these approaches often overlook this population’s diverse needs. This study introduces the “Drive-Fun” innovative intervention (DFI), aimed at enhancing driving skills among this vulnerable population. The intervention was tested in a pilot study including 30 adolescents aged 15–18, comparing three groups: DFI, an educational intervention, and a control group with no treatment. Assessments included a driving simulator, EEG, and Tobii Pro Glasses 2. Evaluation was conducted pre- and post-intervention and at a 3-month follow-up. Results indicated that the DFI group significantly improved in the simulated driving performance, attentional effort, and focused gaze time. The findings underscore that holistic strategies with personalized, comprehensive approaches for adolescents with ADHD are particularly effective in improving driving performance. These outcomes not only affirm the feasibility of the DFI but also highlight the critical role of sensor technologies in accurately measuring and enhancing simulator driving performance in adolescents with ADHD. Outcomes suggest a promising direction for future research and application.

## 1. Introduction

Attention deficit hyperactivity disorder (ADHD) is a neurodevelopmental disorder characterized by inattention and age-inappropriate levels of hyperactivity and impulsivity [1]. There are three presentations of ADHD: predominantly inattentive, predominantly hyperactive-impulsive, and combined (APA, 2013). The prevalence of ADHD ranges from 7% to 20%, with the highest rates occurring among school-age children with a male-to-female ratio of approximately 3:1 [2]. As individuals with ADHD transition from childhood to adulthood, the clinical manifestations of the disorder tend to shift, with behavioral symptoms decreasing and cognitive difficulties persisting [2,3,4].

Road accidents are the most common cause of death among adolescents [5,6,7], and young drivers are involved in more traffic collisions and violations than any other age group [8,9]. Cognitive abilities are a crucial factor in determining the likelihood of risky driving behavior. In particular, inattention and distractibility have been found to account for a quarter of car accidents [10]. Additionally, impaired risk perception and poor judgment and reasoning while driving have been linked to risky behaviors and negative driving outcomes [11]. These higher-order cognitive factors are thought to be the underlying causes of risky driving behaviors. Due to typical immaturity in brain regions governing executive function (EF) [12], combined with their lack of experience, young drivers are susceptible to elevated driving risks. Deficiencies in EF observed in young drivers contribute to their higher collision rates [13].

This risk is even greater for adolescents diagnosed with ADHD, as they have been found to have higher rates of car accidents compared to their peers without ADHD [14]. Driving requires various aspects of attention, including sustained attention, selective attention, divided attention, and alternating attention, which is particularly important in this context [15]. These deficits can lead to difficulties in monitoring the driving environment, processing and responding to information, and switching between tasks, all of which are additional factors increasing the risk of car accidents [16].

Research has demonstrated that adolescents diagnosed with ADHD exhibit inferior driving performance compared to their non-ADHD peers, including a greater frequency of traffic violations, more driving errors, and increased accident rates [17]. In a separate study, Jerome et al. (2006) found that adolescents with ADHD were more likely to be involved in accidents due to, for example, failure to yield the right-of-way, speeding, and distracted driving. Meta-analyses of multiple studies have confirmed that individuals diagnosed with ADHD have a significantly higher likelihood of engaging in risky driving behaviors and experiencing negative outcomes when driving, with estimated risks being 1.23 to 1.88 times higher among this cohort than among individuals without ADHD [16].

Executive functions are critical for driving as they enable drivers to plan, organize, initiate, sustain, and monitor various behaviors in response to both internal and external driving demands [15]. These capabilities are essential for effective goal and strategy management, initiating responses, maintaining working memory, inhibiting distracting or impulsive behaviors, adapting through cognitive flexibility, and solving problems that arise when driving [16]. In adolescents with ADHD, these EFs are often impaired, which is evident in their difficulty with inhibition, working memory, cognitive flexibility, and planning. Such deficits have been thoroughly studied and are particularly concerning as they increase the risk of car accidents among this population [16]. The significant impact of these impairments underscores the critical need for targeted interventions to enhance these cognitive abilities and improve driving safety for adolescents with ADHD [17,18].

Educators and clinicians have developed a variety of interventions aimed at reducing risky driving behaviors among adolescents with ADHD. These interventions range from psychostimulant treatments, which have been shown to enhance driving performance and lower the risk of accidents [18,19,20,21,22], to behavioral interventions and educational programs designed to improve practical driving skills and awareness of driving risks [23,24,25,26,27]. Psychostimulant interventions involve strategic modifications to medication dosing schedules, designed to align with typical activity patterns that require heightened alertness. For instance, for individuals who may need to drive at night, it is recommended to adjust the timing of stimulant administration [28]. Educational interventions for adolescent drivers with ADHD often include targeted driver education programs that emphasize the development of safe driving habits. For example, these programs may incorporate interactive workshops focused on recognizing and managing distractions while driving, understanding the implications of impulsive behavior on road safety, and strategies for maintaining focus over extended periods. Such programs are designed to equip young drivers with the knowledge and skills necessary to navigate the complexities of driving with ADHD effectively [29].

Each of these approaches addresses different aspects of driving challenges faced by adolescents with ADHD. However, adherence to programs and the transient effects of treatments, especially medication, pose ongoing challenges. Indeed, the effectiveness of these interventions varies, highlighting the need for further research to optimize driving outcomes for this population. Thus, previous research has underscored the critical need for holistic strategies with personalized, comprehensive approaches targeted to address the unique challenges faced by adolescents with ADHD in driving contexts. Despite these efforts, no existing method has been fully effective in addressing the significant variability in ADHD symptoms. Each approach has addressed specific challenges, pointing to the necessity for a more holistic solution that can encompass the diverse needs of this cohort.

Attention deficit hyperactivity disorder is characterized by a diverse range of traits and effects that vary from individual to individual. This situation poses a challenge in creating one universal intervention program for all [30]. As Abad-Mas et al. (2013) suggested, interventions for children with ADHD should be tailored to their unique requirements, taking their differences into account. Personalized individualized interactions are crucial in developing effective intervention strategies for adolescents with ADHD.

One such intervention that has shown promise is the teen cognitive functional intervention (Cog-Fun) [31]. This intervention, which was developed and tested among a variety of occupations and with various goals relevant to adolescents with ADHD [32], may be an effective approach when combined with Drive-Fun, a program specifically designed to address driving risks of adolescents with ADHD.

The Cog-Fun intervention model [33] is based on models of cognitive rehabilitation explicitly adapted for individuals with ADHD. The program focuses on developing metacognitive skills and targets the bio-psycho-social barriers to awareness [32]. To address the biological neurocognitive barriers (attention, motivation, and EF deficits), the intervention facilitates learning via the use of structured templates and hierarchical cueing procedures. To address psychological defense mechanisms, Cog-Fun uses a client-centered, strength-based approach and intentional therapeutic relationship techniques [34]. The social barriers to awareness, mainly stigma and lack of knowledge regarding ADHD, are addressed through psychoeducation. Notably, the metacognitive learning process, whereby the client develops adaptive self-awareness, enables the setting of personally meaningful occupational goals. Cog-Fun is effective in treating adolescents with ADHD [32].

Adapting the intervention plan and using the principles of Cog-Fun alongside Drive-Fun—a program developed to reduce risk factors in driving—may be an effective intervention for adolescents with ADHD. The integration of the cognitive effort index (CEI) monitor and Tobii technology during the intervention provided crucial insights. Participants were engaged in feedback sessions during which they reviewed videos of their driving, facilitated by Tobii’s eye-tracking technology. This process allowed them to see where their gaze was focused during driving, offering a reflective learning experience on their spatial awareness and attention distribution. Moreover, the use of the CEI was instrumental in identifying instances of lost focus, periods of optimal attention, and delivering positive feedback on participants’ progress throughout the intervention. By utilizing a combination of guided learning experiences, reflection exercises, and parental involvement, the Drive-Fun intervention program enables participants to engage in interactive sessions within a simulated environment to practice and improve their driving skills. The program aims to promote safe and confident driving behaviors by addressing the unique challenges associated with ADHD and reducing driving risk factors, thus fostering safer and more responsible driving in real-life situations. By utilizing a driving simulator that provides a driving environment mimicking real-world conditions, the program offers a guided learning experience. It incorporates various sources of information, games, and thinking tasks to promote adaptive self-awareness and facilitate the acquisition of management strategies for safe and confident driving. Reflective exercises draw upon participants’ occupational experiences and driving performance within the simulator, while parental involvement ensures ongoing support for the continued practice of learned skills.

The purpose of the current study was to investigate the feasibility of the Drive-Fun intervention program. The Drive-Fun intervention builds upon the teen Cog-Fun program [33], integrating adjustments specifically designed for adolescents with ADHD to enhance their driving skills. To rigorously assess the effectiveness of these interventions, we employed the CEI to measure attentional effort and Tobii Pro Glasses 2 for advanced eye tracking during simulated driving tasks. These objective measures are crucial for providing a clear evaluation of how the interventions impact driving safety and cognitive engagement among participants. Accordingly, our hypotheses are structured as follows:There will be a statistically significant difference in simulator driving grades between the Drive-Fun participants and the control groups, as measured by the 3D-Fahrschule simulator, after the 14-week intervention and again at a three-month follow-up;There will be a statistically significant difference in the middle-range CEI score percentages, as measured by the CEI index, between the Drive-Fun participants and the control groups while driving on a simulator. This difference will be assessed after the 14-week intervention and again at a three-month follow-up;There will be a statistically significant difference in the focused gaze time on the dashboard between the Drive-Fun participants and the control groups, as measured by the Tobii Pro Glasses 2 eye-tracking device. This difference will be assessed after the 14-week intervention and again at a three-month follow-up.

## 2. Materials and Methods

**Participants and demographics.** The study involved a sample of thirty adolescents diagnosed with ADHD, ranging in age from 15 to 18 years. Participants were enlisted via social media networks. To ensure random assignment, a distribution method known as “drawing names from a hat” was employed. Participants’ names were written on individual slips of paper, placed in a hat, and then drawn out one by one to assign them to one of three groups. Each group consisted of ten participants. The groups displayed no significant differences in key indicators, such as age (Group 1: M = 16.5, SD = 0.5; Group 2: M = 17.0, SD = 0.7; Group 3: M = 16.5, SD = 0.5), gender (five boys and five girls in each group), and parental education (M = 17 years, SD = 0.4). All were Caucasian. The inclusion criteria were (1) ADHD diagnosis, confirmed by a child and adolescent psychiatrist, in accordance with DSM-5 criteria; (2) the completion of the Conners 3-Parent Short Form questionnaire by the parents, with a T-score of 60 or higher, indicating potential clinical concerns of ADHD; and (3) no previous driving experience. Exclusion criteria were (1) a chronic primary psychiatric diagnosis and (2) a primary developmental disorder other than ADHD, such as autism spectrum disorder (ASD), intellectual disability, and developmental coordination disorder (DCD). Participants who used amphetamine or methylphenidate were asked not to use them 24 h before evaluation

**Driving simulator.** The 3D-Fahrschule by Besier 3D-Edutainment, Wiesbaden, Germany (Driving School 3D simulator, 2020) recognizes software that trains and examines different aspects of driving in various European countries. Each scenario is awarded points according to the number of mistakes made. The more mistakes made by participants, the higher their score will be. At the end of the driving session, the test results are evaluated on the basis of nine key criteria of unsafe driving behaviors, comprising failure to yield the right of way (3 points), traffic light violations (7 points), failure to stop at stop signs (7 points), speeding (7 points), pedestrian collisions (7 points), incorrect or absent signaling (3 points), improper turns (3 points), vehicular collisions (7 points), and an overall simulator score, which is the cumulative total of all errors. Different scenarios were used in the study phase versus the evaluation pre-study phase. Test participants trained on the 3D-Fahrschule for four minutes before undertaking two test scenarios. We previously validated the simulator by correlating its outcomes with the STISIM Drive, a widely used software tool in driving research and simulation which is commonly used for driving evaluation [35].

**Cognitive Effort Index (CEI):** This index is used for monitoring cognitive effort and provides real-time values from one forehead EEG channel every 10 s. The electrophysiological data is recorded from the NeuroSky EEG MindWave single-channel system (NeuroSky Inc., San Jose, CA, USA), with one frontal electrode and one reference electrode on the earlobe, using a sampling rate of 512 Hz. The NeuroSky device’s utility in measuring cognitive effort and stress effects has been validated across multiple clinical populations and conditions [36,37]. The CEI calculation focuses on delta wave activity (1–4 Hz), and only data filtered to this frequency band are utilized in the analysis, omitting other frequency bands. The sampled data are transferred through a wireless connection to the experimenter’s computer for offline processing. The CEI values are divided into three ranges: 0 ≤ 0.3, 0.3–0.7, and ≥0.7–1.00. The CEI decreases below 0.3 if the task is easy or if the task is too difficult and causes participants to lose attention or even become avoidant. The CEI increases above 0.7 if the task is highly demanding or if it induces an anxious response. The CEI in the middle range (0.3–0.7) indicates effective patient attention [38,39]. These markers are valid and easy to use for real-time monitoring of attention [40].

**Tobii Pro Glasses 2** are a mobile eye-tracking device, appropriate for eye-tracking experiments involving any kind of stimuli in space (TobiiAB, 2015, Danderyd, Sweden). The eye-tracker setup procedure is fast and controlled with a few commands through the Tobii Pro Glasses controller. Participant data are recorded via the controller and can later be accessed through the Tobii Pro Lab software version 1.207. Thus, through Tobii Glasses, we attain and analyze measures of intervention outcomes [41], namely percentage of focused gaze time on the dashboard, percentage of focused gaze time on the pedestrian, and percentage of focused gaze time on traffic light. Before each participant began driving in the simulator, a calibration was conducted to ensure the accuracy of the eye-tracking data. The calibration process was repeated if the quality of the eye tracking was not deemed satisfactory. The Tobii Pro Lab software [42] enables users to accurately determine the time spent on various objects or areas of interest (AOIs). For the driving simulation task, gaze data were meticulously analyzed for three predefined AOIs—the dashboard, traffic lights, and pedestrians—providing detailed insights into the participants’ visual attention during the simulation.

This study, approved by the authors’ university’s ethics committee with approval number 0002841-3, was a controlled pilot study, in which we focused on exploring the practicality and viability of interventions. Participants and their guardians provided informed consent prior to the initiation of the study (they were assured of the study’s ethical adherence, and they understood the study’s scope and purpose). Participants underwent the initial evaluation process before the intervention, which included the following measures: (1) a simulated driving task (3D-Fahrschule by Besier 3D-Edutainment) while being monitored by (2) an eye-tracking device, Tobii Pro Glasses 2, and (3) the CEI. Following the initial evaluation, participants were randomly allocated to one of three groups. Each group received seven individual one-hour sessions at the driving lab at Tel Aviv University and four group 90 min sessions via Zoom.

Following the initial evaluation, all three groups completed their respective interventions. Group 1 was engaged in the Drive-Fun program, a personalized intervention tailored to each individual based on their cognitive and behavioral evaluations. Group 2 received an educational intervention, a standard intervention commonly used in similar studies. As mentioned earlier, Group 3 (control) did not receive any intervention. After 14 weeks, all participants, including Group 3, underwent a second evaluation to assess the impact of the intervention/no intervention. A follow-up evaluation was then conducted, three months later for all groups so that we could assess the maintenance of any observed improvements. Subsequently, Group 3 received a one-time, 90 min guidance session on safe driving after the follow-up evaluation. This comprehensive approach allowed for a thorough examination of the effectiveness and long-term impact of the interventions on driving behavior and skills (see Figure 1).

Data were analyzed using SPSS version 27. Descriptive statistics were calculated for all variables. The intervention effects were tested using a two-way ANOVA with repeated measures, specifically designed to handle three groups across three time-points (i.e., before the intervention, after the intervention, and at a 3-month follow-up). The practical significance of observed effects was assessed using Cohen’s d, and power analyses were conducted.

## 3. Results

In this study we assessed the feasibility of the Drive-Fun intervention program for various driving performance metrics. Specifically, we had five driving performance metrics, which were the outcome variables: (1) the driving simulator score, which aggregates all driving errors listed in the table, where a higher score indicates more infractions committed during the simulation; (2) CEI score in the middle range, which indicates the percentage of time during the full scenario in which the participant exhibits effective attention (see Figure 2 and Figure 3). As can be seen in Figure 2(2a), most CEI values for a typical non-ADHD participant during a driving simulation fall between 0.3 and 0.7, indicating sustained effective attention throughout the scenario, which will be reflected in a greater percentage of CEI scores in the middle range. By contrast, most CEI values for a typical ADHD participant (Figure 2(2b)) fall under 0.3 or above 0.7, indicating non-effective attention throughout the scenario, which will be reflected in a lower percentage of CEI scores in the middle range; (3) focused gaze time on the dashboard, which is the percentage of time participants spend fixating on the dashboard (out of 40 s), rather than on critical driving stimuli, during a set observation period (see Figure 3); (4) focused gaze time on the traffic lights, which is the percentage of time participants spend fixating on the traffic lights (out of 5 s); and (5) focused gaze time on the pedestrian, which is the percentage of time participants spend fixating on the pedestrian (out of 5 s).

Two-way ANOVAs for repeated measures were conducted for each outcome variable, with study groups (Drive-Fun, educational intervention, control) and time (before the intervention, after the intervention, follow-up) as between- and within-subjects’ independent variables, respectively (ANOVA results presented in Table 1).

For the “driving simulator score” variable, significant effects were observed for both time and group, indicating the substantial influence on driving performance. Additionally, the interaction of time and group yielded a significant effect, underscoring their combined impact. Simple slope analysis showed a significant reduction only among the Drive-Fun group but not among the other two groups. Specifically, the driving simulator score was significantly lower both after the intervention and at follow-up, compared to before the intervention, with no significant difference between post-intervention and follow-up (see Figure 4).

In terms of the CEI variable, reflecting EEG-measured attentional effort, time, and group demonstrated significant effects. However, the interaction term did not reach statistical significance. Simple slope analysis showed a significant reduction only among the Drive-Fun group but not among the other two groups. Specifically, the CEI middle-range score was significantly lower both after the intervention and at follow-up, compared to before the intervention, with no significant difference between post-intervention and follow-up. However, the non-significance of the interaction shows that the difference between the groups in the change following the intervention was not significant (see Figure 5).

Regarding the duration of “focused gaze time on the dashboard” while driving, time exhibited significant effects, with the interaction term also proving significant. Simple slope analysis shows a significant reduction only among the Drive-Fun group but not among the other two groups. Specifically, the percentage of “focused gaze time on the dashboard” was significantly lower both after the intervention and at follow-up, compared to before the intervention, with no significant difference between post-intervention and follow-up (see Figure 6).

Regarding the percentage of “focused gaze time on the traffic light” while driving, only the interaction proved to have a significant effect. Simple slope analysis shows a pattern similar to the percentage of focused gaze time on the dashboard, meaning a significant increase only among the Drive-Fun group but not among the other two groups. Specifically, the percentage of “focused gaze time on the traffic light” was significantly higher both post-intervention and at follow-up, compared to before the intervention, with no significant difference between post-intervention and follow-up (see Figure 7).

Regarding the percentage of “focused gaze time on the pedestrian” while driving, no significant effects were found. However, it should be mentioned that simple slope analysis shows a pattern similar to the percentage of focused gaze time on the dashboard and the traffic light, with a significant increase only among the Drive-Fun group but not among the other two groups (see Figure 8).

Overall, the results suggest that the Drive-Fun intervention was the most effective in improving driving performance and attentional effort, compared to both the educational intervention and the no-intervention control groups.

## 4. Discussion

In the current study, we explored the effectiveness of the Drive-Fun intervention program, specifically adapted to enhance the driving capabilities of adolescents diagnosed with ADHD. To determine the efficacy of these interventions, we utilized the CEI for measuring attentional effort, and we employed Tobii Pro Glasses 2 for precise eye tracking during simulated driving tasks. These metrics are essential for an objective evaluation of the intervention’s influence on participants’ driving performance and cognitive abilities.

The overarching goal of this intervention was to foster participants’ awareness and understanding of their multifaceted personal profiles, as well as their ADHD profile, while simultaneously assessing how these profiles impacted their performance in a driving simulator. By enhancing awareness of driving-related challenges and addressing them through the intervention, participants can work toward improved and safer driving outcomes.

The decision to conduct these intervention meetings in a laboratory setting has been supported by research [43,44,45]. Extending the duration of driving simulator practice, introducing challenging stimuli, and providing real-time feedback align with established principles in intervention design [46,47]. These modifications reflect a dynamic and responsive approach to optimizing the intervention’s effectiveness and impact in accordance with the principles of intervention development and implementation [48,49].

Based on the results, the feasibility of the Drive-Fun program in reducing risk factors for car accidents among adolescents with ADHD was supported.

The driving simulator provided a controlled and realistic environment for participants to practice and improve their driving skills [50,51,52]. Through the Drive-Fun intervention, participants learned strategies for correct gaze focus while driving. Such strategies have been demonstrated to enhance driving performance in previous studies [35,53,54] and likely contributed to the observed improvements in driving performance in the current study, such as better attentional allocation and reduced fixation durations on irrelevant stimuli (e.g., the dashboard), as well as increased fixation durations on relevant stimuli (e.g., traffic lights, pedestrians). These findings are consistent with the results of Epstein et al.’s study (2022), in which improvements were also reported in the duration of visual focus on stimuli during driving practice in the simulator. The mediation during driving in the simulator allowed participants to actively apply and practice these strategies, reinforcing the learning process [55].

The CEI served as a critical tool in assessing sustained attention among participants, a vital component of safe driving practices. This metric allowed us not only to evaluate attention span but also to delve into the nuances of attentional effort, as elucidated by Gvion and Shahaf [37]. Our findings revealed that the Drive-Fun group displayed significantly improved attentional effort compared to both the educational program group and the no-intervention group. This noteworthy enhancement in attentional effort among Drive-Fun participants signifies the intervention’s success in bolstering their ability to allocate attention effectively while driving. This allocation of attention, in turn, leads to more focused and attentive driving behavior. The results of this study align with prior research, suggesting that interventions tailored to enhance engagement and awareness during driving practice can positively influence attentional processes [25,56,57,58].

Further investigation into the reasons behind the Drive-Fun group’s improved attentional effort reveals that the intervention likely fostered a state of heightened concentration and interest during driving. Additionally, the Drive-Fun intervention may have encouraged a sense of active participation and self-motivated learning, allowing participants to stay more engaged and immersed in the driving tasks [59].

The adoption of the Tobii eye-tracking system in this study represents an innovative approach to analyzing visual attention dynamics during driving. By emphasizing the criticality of precise gaze focus and minimizing distractions, the Drive-Fun program equips participants with the necessary strategies to refine their gaze patterns, in line with Parsons [60], who highlighted the pivotal role of gaze control in maintaining attention. The group’s improved gaze behavior, characterized by reduced fixations on irrelevant stimuli, echoes the positive outcomes of interventions aimed at optimizing gaze control for enhanced environmental scanning capabilities. In our analysis, we specifically focused on three predetermined AOIs: the dashboard, a pedestrian, and a traffic light. We chose the dashboard as an AOI given its characteristics as a static stimulus, a decision supported by previous research suggesting that individuals with ADHD are prone to fixate on static stimuli, such as the dashboard, potentially to the detriment of their awareness of more dynamic and immediately relevant driving cues [41]. Such fixation behavior is particularly significant as it could divert attention away from critical dynamic elements, such as pedestrians and traffic lights, which are vital for safe driving.

The implications of the tendency to focus on static stimuli are profound. Although static elements, such as the dashboard, provide important driving information, the excessive focus on these elements at the expense of dynamic elements can compromise the driver’s ability to react to sudden changes in the traffic environment. This behavior could increase the risk of accidents, especially in complex driving scenarios where attention to dynamic stimuli is crucial for timely and appropriate responses. Therefore, understanding the fixation patterns on static versus dynamic stimuli in individuals with ADHD not only adds to our knowledge of attentional processing in this population but also underscores the need for tailored driving assessments and training programs that address these specific challenges.

It is important to highlight that the educational program control group also exhibited some improvements in simulator performance and attentional effort, albeit to a lesser degree than did the Drive-Fun group. This finding suggests that there might be shared elements in these programs, contributing to their efficacy in mitigating risk factors associated with car accidents among adolescents with ADHD. Going forward, researchers could delve into the specific components of these programs that have the most significant impact.

## 5. Limitations

The study sample size was relatively small; as such, the results should be interpreted with caution. This study was conducted as a feasibility study to assess the acceptability and practicality of the Drive-Fun intervention. The insights gained from this study will inform refinements to the intervention and research protocol for a larger randomized controlled trial (RCT).

## 6. Conclusions

The findings of this study highlight the feasibility of Drive-Fun as an intervention program to improve driving performance, attentional effort, and eye movements among adolescents with ADHD. To learn more about the effectiveness of this intervention in improving learning strategies incorporated into the intervention, particularly those related to correct gaze focus and attentional allocation, a larger study should be completed. The use of the driving simulator, the CEI, and the Tobii eye-tracking system may provide valuable insights into participants’ driving skills and cognitive processes, allowing for a comprehensive evaluation of the feasibility of the intervention.

Based on these results, we are currently increasing the sample size so as to further explore the specific mechanisms underlying the observed changes and investigate the long-term effects of Drive-Fun.

## Figures and Tables

**Figure 1 sensors-24-03319-f001:**
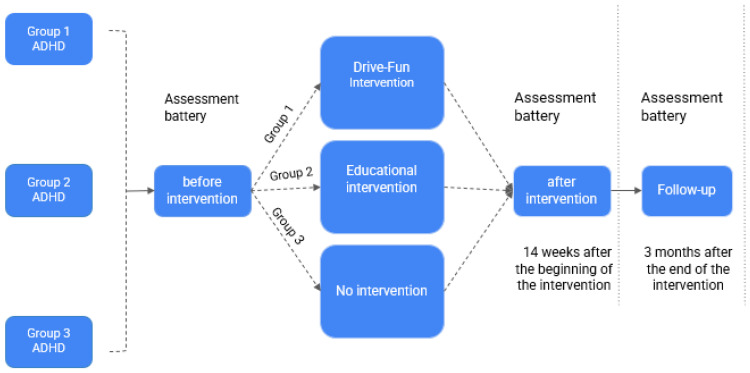
Flow diagram of study design.

**Figure 2 sensors-24-03319-f002:**
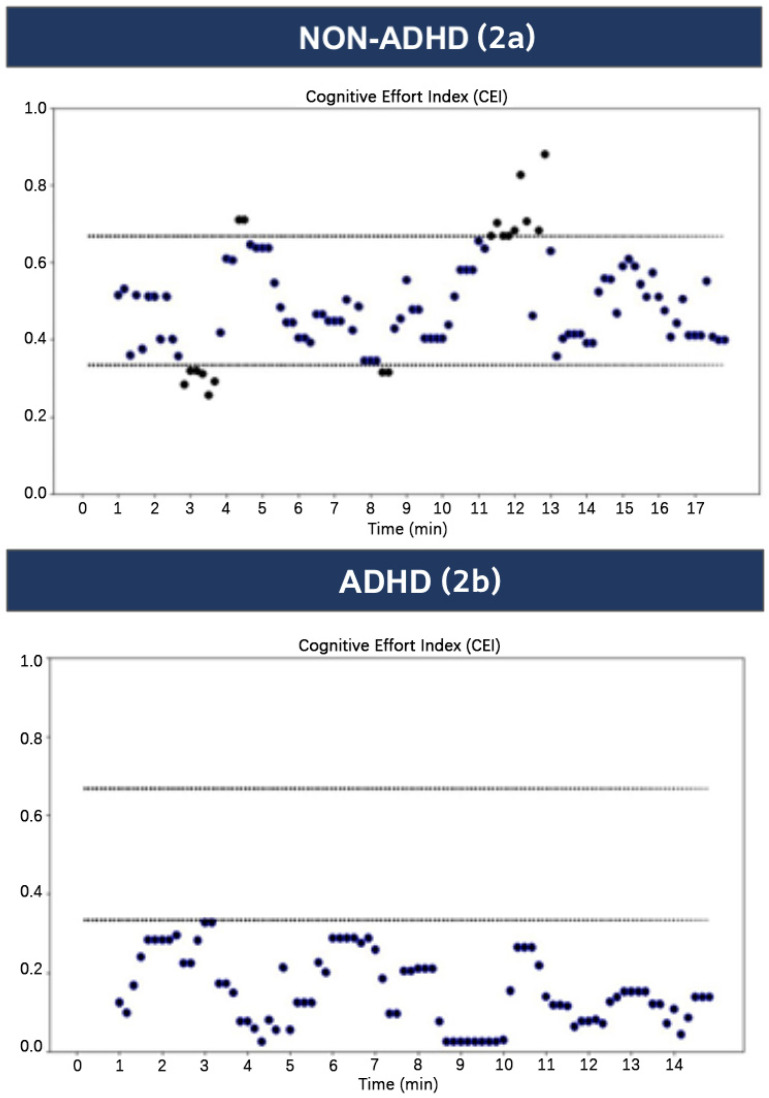
CEI distribution during driving simulation for non-ADHD and ADHD participants. The first graph (**2a**) shows the CEI for a typical non-ADHD participant during a driving simulation. The second graph (**2b**) shows the CEI for a participant with ADHD.

**Figure 3 sensors-24-03319-f003:**
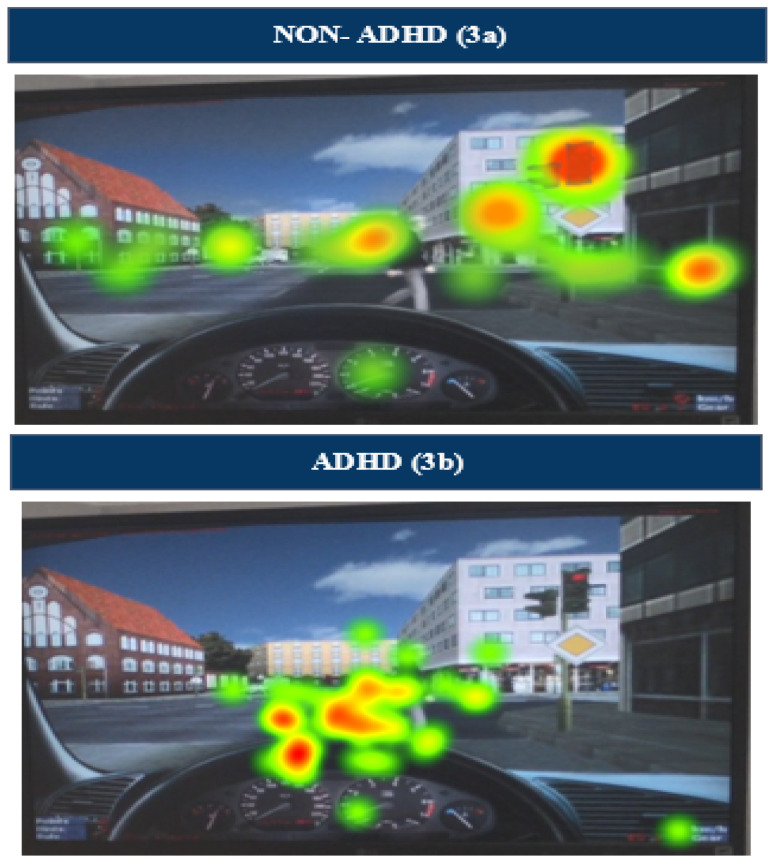
These heat map images illustrate the differences in gaze patterns between adolescents with and without ADHD, during a driving simulation. The first image (**3a**) shows a typical non-ADHD participant focusing primarily on critical stimuli, such as the pedestrian at the crossing signal (red light), while the second image (**3b**) highlights participants with ADHD tending to focus more on the dashboard. This visual comparison underscores the importance of the dashboard fixation metric, which significantly differs between the two groups, reflecting the distinct attentional behaviors influenced by ADHD.

**Figure 4 sensors-24-03319-f004:**
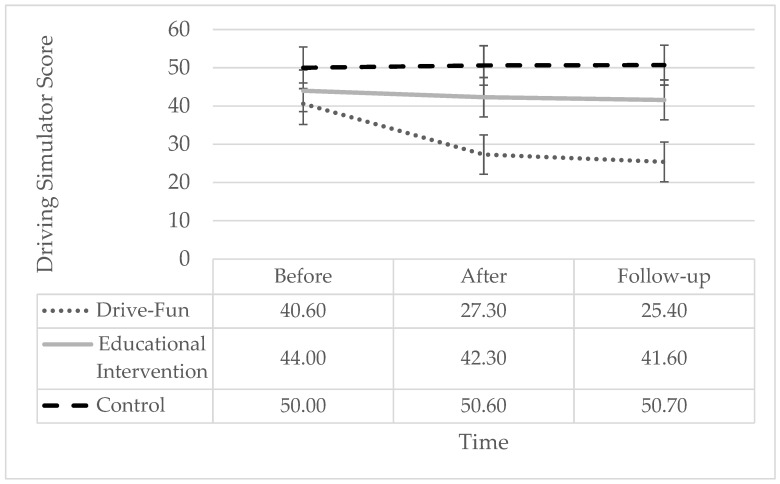
Driving simulator score by study groups and time. The error bars represent the standard error. Simple effect analysis shows a significant reduction after the intervention (*p* < 0.001) and at follow-up (*p* < 0.001), compared to before the intervention, only among the Drive-Fun group. There are no other significant differences (*p* < 0.05).

**Figure 5 sensors-24-03319-f005:**
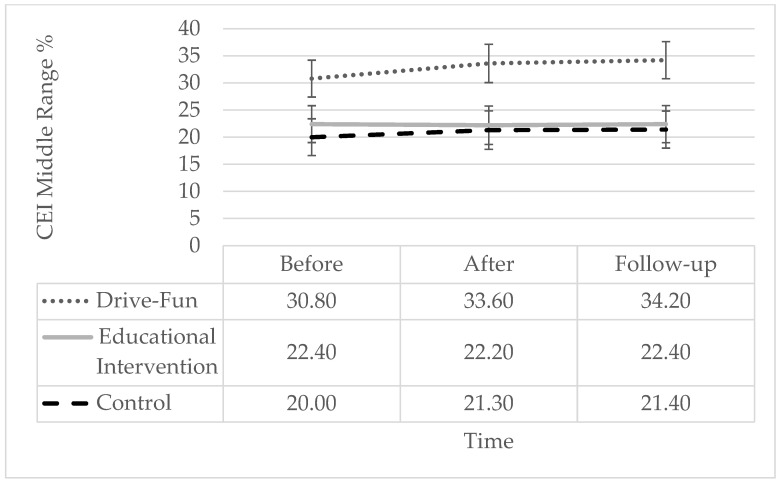
CEI middle-range score by study groups and time. The error bars represent the standard error. Simple effect analysis shows a significant reduction after the intervention (*p* = 0.002) and at follow-up (*p* < 0.001), compared to before the intervention, only among the Drive-Fun group. There are no other significant differences (*p* < 0.05).

**Figure 6 sensors-24-03319-f006:**
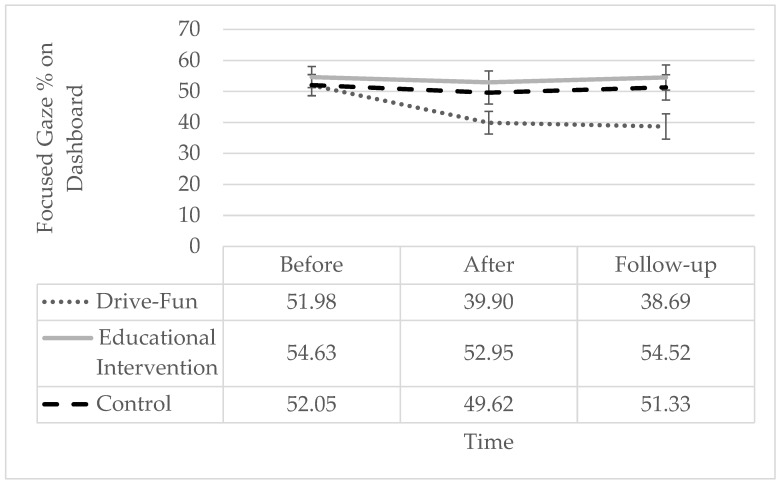
Percentage of focused gaze time on the dashboard by study groups and time. The error bars represent the standard error. Simple effect analysis shows a significant reduction post-intervention (*p* = 0.001) and at follow-up (*p* < 0.001), compared to before the intervention, only among the Drive-Fun group. There are no other significant differences (*p* < 0.05).

**Figure 7 sensors-24-03319-f007:**
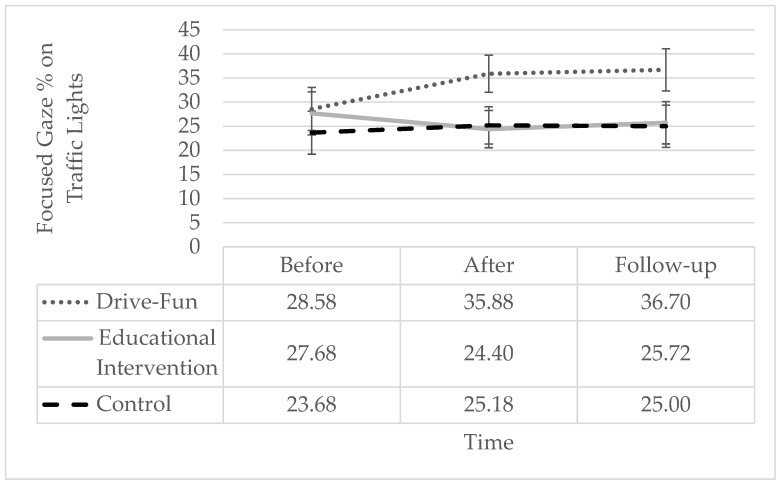
Percentage of focused gaze time on traffic lights by study groups and time. The error bars represent the standard error. Simple effect analysis shows a significant reduction post-intervention (*p* = 0.004) and at follow-up (*p* = 0.001), compared to before the intervention, only among the Drive-Fun group. There are no other significant differences (*p* < 0.05).

**Figure 8 sensors-24-03319-f008:**
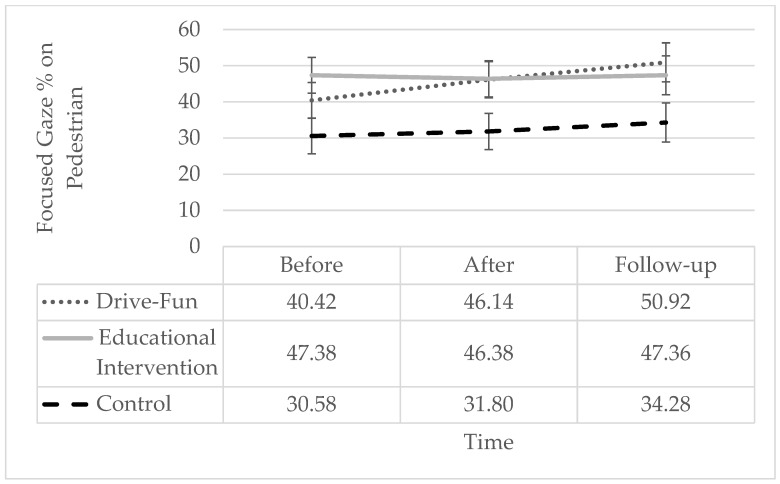
Percentage of focused gaze time on the pedestrian by study groups and time. The error bars represent the standard error. Simple effect analysis shows a significant reduction after the intervention (*p* = 0.004) and at follow-up (*p* = 0.005), compared to before the intervention, only among the Drive-Fun group. There are no other significant differences (*p* < 0.05).

**Table 1 sensors-24-03319-t001:** Statistical analysis results for study groups and time effects (N = 30).

		F	Df	*p*	Cohen	Power
Driving simulator score	Time	12.02	2.26	<0.001	0.48	0.99
	Group	3.56	2.27	0.043	0.21	0.61
	Time × Group	7.28	4.52	<0.001	0.36	0.99
CEI middle range value (%)	Time	5.32	2.26	0.012	0.29	0.79
	Group	3.63	2.27	0.040	0.21	0.62
	Time × Group	1.93	4.52	0.119	0.13	0.54
Focused gaze time on dashboard (%)	Time	5.26	2.26	0.012	0.29	0.79
	Group	2.66	2.27	0.088	0.17	0.48
	Time × Group	3.07	4.52	0.024	0.19	0.77
Focused gaze time on traffic light (%)	Time	1.89	2.26	0.172	0.13	0.36
	Group	1.46	2.27	0.251	0.10	0.28
	Time × Group	2.70	4.52	0.040	0.17	0.71
Focused gaze time on pedestrian (%)	Time	3.17	2.26	0.059	0.20	0.56
	Group	2.90	2.27	0.072	0.18	0.52
	Time × Group	1.93	4.52	0.120	0.13	0.54

## Data Availability

The raw data supporting the conclusions of this article will be made available by the authors on request.

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
