# Peer review of "Reducing Driving Risk Factors in Adolescents with Attention Deficit Hyperactivity Disorder (ADHD): Insights from EEG and Eye-Tracking Analysis"

_sensors, 2024, doi:10.3390/s24113319_

Round 1
Reviewer 1 Report
Comments and Suggestions for Authors
The topic of this work is interesting, but the current version needs major corrections to be more readable for the Sensors journal audiences.
The authors proposed a method for reducing driving risks in 30 adolescents, 15-30 year olds with ADHD using the “Drive-Fun” intervention.
The current version is written as a scientific report not as a research paper, for that the authors should be carefully addressed the following corrections in the revised version:
1- The abstract section should be written as one paragraph and improved the current version to include the following points
- What are the research gaps,
- What are the main contributions of this study,
- What are the main conclusions of the work?
2- In the Introduction section summarize the previous work to provide the significance of your method.
3- Section "2. Research Hypothesis" is too short and it can be included in the Introduction Section.
4- The method Section should be improved and written in a more professional way to be more readable and provide an overview of the contribution of this work.
5- In the Abstract section "Participants 15-30 -year-old with ADHD" and in the Method section "Participants: The study involved a sample of thirty adolescents diagnosed with ADHD, 152 ranging in age from 15 to 18 years."?
which age group is correct?
6- Many sections will lead to loss the main idea of the proposed work, please try reorganizing the whole section using the Sensors journal guide.
Comments on the Quality of English Language
There are some typos and grammar mistakes, the whole paper should be reviewed by an English native speaker.
Reviewer 2 Report
Comments and Suggestions for Authors
Anat Keren and colleagues report from a pilot study addressing the effects of drive-fun intervention on adolescents with ADHD on driving performance in a driving simulator, together with EEG and eye movement properties.
30 adolescent participants with ADHD were divided into three experimental groups: drive-fun intervention, educational intervention, and control - no intervention at all.
Data were obtained at three different times: Before and after the intervention (or no intervention in the control group) and follow-up. The interval between first and second experiment was 14 weeks (not explicitly stated in the text), no information is given about the delay of the follow-up experiment.
Unfortunately, the authors do not show any data, only the results of statistical testing (two-factorial ANOVA, effect of group, time and interaction) are given.
Firstly, the values of the parameters from the driving simulator (number of mistakes) should be given.
Secondly, it should be explained how exactly CEI was determined? Typical examples of the obtained EEG should be shown, eventually the power of alpha, beta, and gamma band. To me, the NeuroSky device is very reminiscent to a famous toy from Mattel called Mind Flex. It might be questionable whether it can be used for scientific experiments in academia.
Thirdly, eye tracking data should also be shown: number of saccades per interval as well as amplitudes and peak velocity of saccades, stability of fixation, number of micro-saccades and finally pupil size over time would be important for the reader.
Without showing the experimental data (i.e. parameters of driving simulator, recorded EEG wave forms, and results from eye tracking) for typical subjects as well as group data, it is not possible to estimate the scientific quality of the manuscript
Minor comment
The following references are not included in reference list
Wu 2019
Parsons 2015
Round 2
Reviewer 1 Report
Comments and Suggestions for Authors
The authors addressed my comments correctly, only Figure 2 needs to improve.
Reviewer 2 Report
Comments and Suggestions for Authors
The authors performed a careful revision of their initial manuscript. I do not have additional comments